# Early Arterial Intimal Thickening and Plaque Is Related with Treatment Regime and Cardiovascular Disease Risk Factors in Young Adults Following Childhood Hematopoietic Stem Cell Transplantation

**DOI:** 10.3390/jcm9072208

**Published:** 2020-07-13

**Authors:** Johnny K. M. Sundholm, Anu Suominen, Taisto Sarkola, Kirsi Jahnukainen

**Affiliations:** 1Division of Cardiology, Children’s Hospital, University of Helsinki and Helsinki University Hospital, 00029 Helsinki, Finland; taisto.sarkola@helsinki.fi; 2Division of Hematology—Oncology and Stem Cell Transplantation, Children’s Hospital, University of Helsinki and Helsinki University Hospital, 00029 Helsinki, Finland; anu.suominen@fimnet.fi (A.S.); kirsi.jahnukainen@ki.se (K.J.); 3Minerva Foundation Institute for Medical Research, 00290 Helsinki, Finland; 4Department of Women’s and Children’s Health, Karolinska Institute and University Hospital, 17164 Stockholm, Sweden

**Keywords:** early vascular ageing, atherosclerosis, radiotherapy, hematopoietic stem cell transplantation, very-high resolution ultrasound

## Abstract

The long-term vascular effects following childhood hematopoietic stem cell transplantation (HSCT) are not well characterized. We compared arterial wall morphology and function using very-high resolution ultrasound (25–55 MHz) in 62 patients following autologous (*n* = 19) or allogenic (*n* = 43) HSCT for childhood malignancies and hematological disease (median age 25.9 years, IQR 21.1–30.1; median follow-up time 17.5 years IQR 14.1–23.0) with an age matched healthy control group (*n* = 44). Intima-media thickness of carotid (CIMT 0.49 ± 0.11 vs. 0.42 ± 0.06 mm, *p* < 0.001), brachial, femoral, radial arteries, and local carotid stiffness, but not adventitial thickness, were increased (*p* < 0.001). Diffuse intimal thickening (>0.06 mm) of femoral or radial arteries (*n* = 17) and subclinical carotid or femoral plaques (*n* = 18) were more common (*p* < 0.001). Radiation predicted plaques (*p* < 0.001) and local carotid stiffness (*p* < 0.001), but not intimal thickening. CIMT was predicted by age, BMI >30 kg/m^2^, hsCRP >2.5 mg/L, hypertension, HbA1c > 42 mmol/L, and cumulative anthracycline >150 mg/m^2^. Cumulative metabolic syndrome criteria and cardiovascular disease (CVD) risk factors were more common among HSCT and related with CIMT (*p* < 0.001), but CIMT was similar among controls and HSCT without CVD risk factors. Long-term childhood HSCT survivors show early arterial aging related with radiation, metabolic, and CVD risk factors. Prevention of risk factors could potentially decelerate early arterial wall thickening.

## 1. Introduction

More children are treated with hematopoietic stem cell transplantation (HSCT) for malignant and non-malignant diseases [1]. Survivors are prone to both disease, pretreatment and treatment regime related long term morbidities. Cardiovascular disease (CVD) risk factors are prevalent among childhood cancer survivors treated with HSCT [2,3,4,5,6,7].

Although the increased prevalence of CVD risk factors following HSCT has been described more thoroughly among adults, there is still lack of information regarding surrogate markers of subclinical CVD including arterial stiffness and arterial wall abnormalities in long-term survivors following pediatric HSCT [8]. The interrelations between intima-media thickness (IMT), pulse wave velocity (PWV) and CVD risk factors have not previously been assessed in more detail in this population. The implementation of non-invasive vascular very-high resolution ultrasound (VHRU, 25–55 MHz) has made it possible to investigate the vascular wall in more detail (axial pixel resolution 0.015–0.040 mm) [9]. Peripheral arterial vascular wall layer thickness is related with CVD events, but the relationship is not as well established as for carotid IMT and plaque [10,11]. We have recently histologically verified the VHRU derived four-line pattern of the arterial far wall as a non-invasive method to assess diffuse intimal thickening simultaneously and separately from medial and adventitial layer thickness [12].

The aims of this study were (1) to assess the prevalence of metabolic CVD risk factors and their surrogate subclinical CVD markers, including arterial layer thickness, plaque, and stiffness, among young adult childhood HSCT survivors with healthy age-matched controls and (2) to assess interrelations between metabolic and CVD risk factors, treatment regimens, and surrogate subclinical CVD markers.

## 2. Study Subjects and Methods

This is a single-center cross-sectional case-control study consisting of two cohorts of HSCT survivors (*n* = 62). In the first cohort, all male survivors treated between 1980 and 2010 at the age of up to 17 years with myeloablative allogeneic HSCT and who on 1 January 2017 were at least 18 years of age were eligible for the study (*n* = 85). Twenty-seven patients had missing contact information or were non-responders, 6 patients declined participation, and 2 patients were excluded as they were unable to give consent. Another 7 subjects were not eligible for the vascular assessment providing a final sample of 43 male subjects. Non-participants did not differ significantly from participants regarding background data.

The second cohort consisted of all high-risk neuroblastoma survivors (*n* = 23) born 1996 or earlier and treated with autologous HSCT in Finland between 1980 and 2000. The cohort also included one high-risk retinoblastoma patient with same treatment regime. Twenty patients consented but one patient died prior to study initiation giving a final sample of 19, the cohort has been further described in Vatanen et al., 2015 [13].

Forty-four healthy controls without history of chronic disease, symptoms or current medication were recruited among hospital personnel, students and teenage children among hospital employees. The controls were age matched as a group with HSCT cases, and the sex distribution between the study groups was similar. The study protocol conforms to the ethical guidelines of the 1975 Declaration of Helsinki, and it was approved by the local ethics board (Helsinki University Hospital, Ethics Committee for gynecology and obstetrics, pediatrics and psychiatry HUS/1742/2016 and HUS 183/13/03/03/2010). Written informed consent was obtained from all participants at enrollment.

Information on diagnosis, treatment, and conditioning regimes was gathered from hospital records including exposure to irradiation (TBI and local testicular and CNS irradiation). Exposure to alkylating agents was calculated as a cumulative cyclophosphamide equivalent dose (CED, mg/m^2^) [14]. Cumulative exposure to anthracyclines (mg/m^2^) was calculated with the following conversion factors: Doxorubicin 1.0, Daunomycin 0.833, Idarubicin 5.0, Epirubicin 0.67, and Mitoxantrone 4.0. Acute graft versus host disease (aGVHD) and its gradus were assessed from patient records, whereas chronic GVHD (cGVHD) was prospectively assessed [15].

### 2.1. Anthropometrics, Body Composition, Blood Pressure, and Blood Work

An electronic stadiometer and scale (Seca gmbh & co. kg, Hamburg, Germany) was used to measure height to the nearest 0.1 cm and weight to the nearest 0.1 kg with light clothing and without shoes (Seca 770, Seca gmbh & co. kg, Hamburg, Germany). Hip and waist circumference were measured with a tape measure to the nearest 0.5 cm. Body surface area (BSA) was calculated using Mosteller formula [16]. Body composition was measured by dual X-ray absorptiometry (DXA) using a Lunar Prodigy Advance fan beam scanner (GE Medical Systems, Madison, WI, USA) and analyzed using Prodigy enCORE software version 16.10.15).

Blood pressure (BP) was assessed using three right brachial oscillometric measurements (Omron, M6W BP monitor–device, Omron HealthCare Europe, Hoofddorp, The Netherlands) at rest in the sitting position. The mean of the two lowest readings were used in analyses.

Laboratory tests where assessed from venous blood samples collected between 8 and 10 am following overnight fasting and analyzed according to standard procedures at the Helsinki University Hospital laboratory.

Current medications, diagnoses, and smoking history was assessed by questionnaire. Hypertension was defined as either a systolic blood pressure >140 mmHg, diastolic blood pressure >90 mmHg, or use of anti-hypertensive medication. Diabetes was defined as a fasting glucose ≥6.9 mmol/L, glycated hemoglobin (HbA1c) ≥ 48 mmol/mol, or use of glucose lowering medication. Hypercholesterolemia was defined as LDL >3.0 mmol/L or treatment with cholesterol lowering medication. Metabolic syndrome was classified according to AHA guidelines, fulfilling at least three out of following five criteria: (1) Waist circumference >102 cm in men and 88 cm in women; (2) Triglycerides > 1.7 mmol/L or treatment for hypertriglyceridemia; (3) High-density lipoproteins <1.03 mmol/L in men and 1.3 mmol/L in women; (4) Systolic blood pressure > 130 mmHg, diastolic blood pressure >85 mmHg, or antihypertensive medication; and (5) Fasting plasma glucose > 6.1 mmol/L or glucose lowering medication [17]. Waist circumference was not available for HSCT cohort 2.

### 2.2. Vascular Ultrasound

Vascular VHRU moving cine-clips were acquired from carotid, radial and femoral arteries using Vevo MD (VisualSonics, 2016, Toronto, ON, Canada) equipped with UHF22, UHF48, and UHF70 transducers (center frequencies 15 MHz, 30 MHz, and 50 MHz respectively, cohort 1) or Vevo 770 (VisualSonics, 2005, Toronto, ON, Canada, cohort 2) equipped with 25, 35, and 55 MHz transducers. All images were acquired by one experienced investigator (TS) as previously reported [12]. Previous intra-arterial lines in the vessels scanned were prospectively recorded by interview and inspection of skin scar. Measurements were performed offline with the operator blinded to subject characteristics from the far wall in end-diastole with manual electronic calipers [12,15]. The intra-observer coefficients of variations (CV) for different arterial VHRU measurements were 1.2–2.9% for LD, 6.9–9.8% for IMT, and 7.6–28.6% for AT, and inter-observer CVs were 1.5–4.6% for LD, 6.0–10.4% for IMT, and 5.9–20.5% for AT [18].

We have recently histologically validated the far wall four-line pattern in the VHRU arterial image and shown that the intima layer thickness can independently and separately be accurately quantified using the leading edge-to-leading edge technique in arteries with a histological intima layer thickness of more than 0.06 mm [12]. Due to the low prevalence of intimal thickening in the sample, we recorded intimal thickening as a binominal variable defined as the presence of a four-line pattern in the muscular artery far wall. Cohen’s Kappa for intra- and inter-observer agreements were κ = 1.00 and κ = 0.85, respectively.

Carotid artery lumen diameter was further measured during peak systole, and the local carotid artery β-stiffness index (CBSI) and distensibility coefficient (CDC) were calculated as:(1)$$CDC=1000\times \frac{\left(CCALAS-CCALAD\right)÷CCALAD}{\left(BPS-BPD\right)}$$
(2)$$CBSI=\frac{\mathrm{lna}\left(\frac{BPS}{BPD}\right)}{\left(CCALDS-CCALDD\right)/CCALDD}$$
where *BPD* = diastolic blood pressure, *BPS* = systolic blood pressure, *CCALAD* = Carotid artery diastolic lumen area, *CCALAS* = carotid artery systolic lumen area, *CCALDD* = carotid artery diastolic lumen diameter, and *CCALDS* = carotid artery systolic diameter.

### 2.3. Plaque-Screening

The carotid artery was screened from the common carotid artery prior to the bifurcation, throughout the bulb and the proximal parts of the internal and external carotid arteries. The femoral artery was screened from the common femoral artery throughout the bifurcation and the proximal parts of both the deep and superficial femoral arteries. Plaques were defined in accordance with the Mannheim consensus as a focal thickening of the vascular wall fulfilling at least one of three criteria: (1) maximum IMT >1.5 mm, (2) IMT increase of 0.5 mm, or (3) 50% increase in IMT compared with the surrounding vascular wall. Plaque burden was further classified as single or multiple plaques [19]. Cohen’s Kappa for intra- and inter-observer agreement on plaque presence were κ = 1.00 and κ = 0.90, respectively, and for multiple plaque presence κ = 0.95 and κ = 0.86, respectively.

### 2.4. Pulse Wave Velocity

Regional PWV was measured in HSCT cohort 1 by one investigator (TS) using mechanosensors (Complior Analyse, Alam Medical, Saint-Quentin-Fallavier, France) with transit time recorded simultaneously at carotid, radial, and femoral arteries. The direct distance between the recording sites was measured with a tape measure to the nearest 0.1 cm. The direct distance between carotid and femoral sites was multiplied by 0.8. The CV for repeat measures were 1.2% for carotid-radial and 3.6% for carotid-femoral PWV.

### 2.5. Data Analysis

Data are presented as mean and standard deviation, median and interquartile range, or proportions, as appropriate. Continuous variables were assessed for normal distribution using Shapiro–Wilks test. Group-wise comparisons were done using independent Student’s *t*-test for normally distributed variables, Mann–Whitney U-test for non-normally distributed variables, and Fisher’s exact test for binominal variables. Prevalence of CVD risk factors and carotid plaques were compared to a population reference using Chi-square goodness of fit.

Group differences between vascular parameters were further compared using ANCOVA adjusting for age and BSA. PWV models were further adjusted for heart rate and mean arterial pressure. Comparison between multiple groups after sample stratification was assessed using Fisher–Freeman–Halton Exact-test, post hoc pair wise Fisher’s exact tests with Bonferroni adjusted significance levels for binominal variables, and for continuous variables with ANOVA post hoc Bonferroni adjusted pair wise *t*-tests.

Multiple linear regression and ANCOVA models were used to assess direct effects of treatment, i.e., TBI, CED, and anthracyclines on carotid IMT adjusting for BMI, hypertension, glucose, blood cholesterols, and smoking with non-significant parameters excluded from the final model. Models were built using a bootstrapped sample (1000 repeats) and validated using the study sample. Interaction effects were assessed, as appropriate. Multicollinearity was investigated using variance influence factor (VIF), with VIF not exceeding 2.5 for any parameter in any of the models. Normality and homoscedasticity of residuals was assessed, as appropriate.

Intra- and inter-observer agreements were assessed using CV for continuous variables and Cohen’s Kappa (κ) for binominal variables.

## 3. Results

### 3.1. HSCT Patient and Control Characteristics

Mean age at diagnosis was 8.1 years (IQR 3.0–12.0 years) and follow-up time was 17.5 years (IQR 14.1–23.0 years). The primary diagnoses were acute lymphoid leukemia (*n* = 26), high-risk neuroblastoma (*n* = 18), severe aplastic anemia (*n* = 6), acute myeloid leukemia (*n* = 5), chronic granulomatous disease (*n* = 2), myelodysplastic syndrome (*n* = 2), high risk retinoblastoma (*n* = 1), dyserythropoietic anemia (*n* = 1), and Chediak–Higashi syndrome (*n* = 1). Forty-six patients (74%) had TBI, out of which 5 (8%) had additional gonadal irradiation and 7 (11%) additional CNS-irradiation. Forty-two (68%) patients had cyclophosphamide (median cumulative dose 5541 mg/m^2^, range 2000–23,183 mg/m^2^), whereas 41 (66%) had anthracyclines (median cumulative dose among receivers 165 mg/m^2^, range 40–500 mg/m^2^), with radiated patients receiving more anthracyclines than non-radiated patients. Acute graft-versus-host disease (aGVHD) was recorded in 33 patients (53%), with severe GVHD (grade 3–4) in 10 patients (16%). Chronic GVHD (cGVHD) was recorded and graded in 10 patients (16%) at the study visit with five patients showing moderate to severe cGVHD (8%). Further background data is presented in Appendix A.

Data on anthropometrics and CVD risk factors in HSCT survivors and controls at follow-up are presented in Table 1. HSCT survivors were shorter and lighter than controls, and there was no difference in BMI or prevalence of overweight or obesity. There were no statistically significant differences in systolic or diastolic blood pressures between HSCT survivor and control groups, but heart rate was higher among HSCT survivors.

Prevalence of overweight among HSCT patients was lower compared to a Finnish reference population (29.0 vs. 58.8%, *p* = 0.001), whereas the difference in the prevalence of obesity was borderline insignificant (11.3 vs. 21.5%, *p* = 0.051) [20]. Compared to a Finnish reference population of similar age, HSCT patients had a higher prevalence of hypertension (33.9 vs. 12.8%, *p* < 0.001), similar prevalence of hypercholesterolemia (33.9 vs. 28.6%, *p* = 0.353), and a higher prevalence of diabetes (8.1 vs. 2.3%, *p* = 0.002) [21].

The prevalence of carotid artery plaques was significantly higher among HSCT patients than in a previously reported Norwegian population of similar age (24.2 vs. 2.8%, *p* < 0.001) [21]. The presence of arterial intimal thickening was also significantly more prevalent among HSCT patients than we have previously reported among subjects aged 25–50 years with significant CVR burden (31.0 vs. 4.5%, *p* < 0.001) [12].

### 3.2. Arterial CVD Surrogate Marker Differences between HSCT Survivors and Controls

Sample images of arterial plaques and arterial wall layers among HSCT survivors are illustrated in Figure 1.

Unadjusted and adjusted vascular dimension data are presented in Table 2. HSCT survivors had smaller lumen diameters in radial and femoral arteries compared to controls and thicker IMT in carotid, brachial, and femoral arteries, with no consistent difference in adventitia thickness. The difference in LD disappeared in all arteries except for the femoral arteries when adjusting for age and BSA, whereas the difference in IMT between HSCT survivors and controls was exaggerated and statistically significant for all evaluated arteries, including the radial artery.

The increase in CIMT with age was steeper in independent linear regression models among HSCT survivors (β = 8.8 µm/year, CI95% 5.1–12.5 µm/year) compared to controls (β = 3.4 µm/year, CI95% 1.0–5.7 µm/year), seen as a significant interaction term in a multiple linear regression model (age * HSCT β = 5.5 µm/year CI95% 0.9–10.1 µm/year, *p* = 0.020, Figure 2, Appendix A).

Three HSCT survivors and one control were excluded from the intimal-thickening analysis due to a history of previous intra-arterial catheters. Intimal thickening in any artery was more frequent among HSCT than controls (18/62 vs. 1/44, *p* < 0.001; Appendix A) with no significant difference between non-radiated and radiated survivors (4/15 vs. 14/43, *p* = 0.756). HSCT survivors with intimal thickening were significantly older than HSCT without intimal thickening (29.5 ± 8.0 years vs. 24.7 ± 5.6 years, *p* = 0.01, Appendix A).

Carotid and femoral plaques were more frequent among HSCT than controls (18/62 vs. 2/44, *p* = 0.001; Appendix A). Twelve out of 15 survivors with plaques had multiple plaques, whereas none of the controls had multiple plaques (*p* = 0.005).

Carotid beta stiffness index (CBSI) was higher and carotid distensibility coefficient (CDC) lower in HSCT compared to controls (adjusted *p* < 0.001). However, there was no statistically significant differences between HSCT survivors and controls in carotid-radial or carotid-femoral PWVs (Table 2).

### 3.3. Arterial CVD Surrogate Markers in Relation to Treatment Regimens and CVD Risk Factors among HSCT

There were no significant differences in arterial wall dimensions, PWV, intimal thickening, or arterial plaques in HSCT exposed to TBI compared with non-exposed (Appendix A).

HSCT survivors with local additional boost radiation therapy of the gonads had higher prevalence of femoral artery plaques compared to HSCT survivors treated with TBI only (3/5 vs. 4/41 *p* = 0.020) as well as compared to non-radiated subjects TBI (3/5 vs. 1/16, *p* = 0.028, Appendix A). Local additional radiation therapy of the CNS area was not, however, associated with more carotid plaques (1/7 vs.13/39 *p* = 0.413, Appendix A). Patients with plaques were slightly older than patients without plaques (30.0 years vs. 24.5 years, *p* = 0.003), but plaque presence was unrelated to traditional CVD risk factors in our sample (results not shown).

There were minor differences between allogenic and autologous HSCT survivors (Appendix A), but only femoral IMT remained statistically significantly thicker in allogenic HSCT after adjusting for age, BSA, and sex. Local carotid artery stiffness measures were higher in allogenic HSCT.

In ANCOVA models, age, BMI, HbA1c, hypertension, hs-CRP, and cumulative anthracycline dose were all independently related with CIMT among HSCT recipients (Table 3). There was also a dose related association between CIMT and cumulative anthracycline exposure. Similarly, there was a positive trend between CIMT and cumulative number of CVD risk factor and metabolic syndrome criteria (Figure 3A,B). No difference was, however, seen between HSCT without metabolic or CVD risk factors and healthy controls.

Both CBSI and CDC were significantly predicted by age, low-density lipoprotein, and TBI exposure (Appendix A), but no association with PWV was found. Radial, brachial, and femoral artery IMT were predicted by age but not by CVD risk factors or treatment regimens (results not shown). There were no statistically significant associations between vascular parameters and previous aGVHD, aGVHD grade, or active cGVHD or cGVHD severity (results not shown).

## 4. Discussion

We compared arterial wall morphology and stiffness in a cohort of young adult HSCT survivors of childhood malignancies and hematological diseases with healthy controls and further assessed relations between vascular parameters, treatment regimens, and CVD risk factors. We report, similar to previous reports, reduced body size, higher heart rate, and increased prevalence of subclinical arterial plaques among HSCT long-term survivors [3,4,22]. Overall, among HSCT patients, CVD risk factors were more common than in the average population, and arterial walls were thicker, stiffer, and presented more plaques and intimal thickening compared to healthy controls without CVD risk factors, consistent with early vascular ageing [13,23].

Local arterial stiffening and plaques were related with radiation therapy, and femoral artery plaques were pronounced in patients with additional gonadal irradiation, as previously reported [24,25,26]. A similar relationship was not seen between additional CNS irradiation and carotid artery plaque likely explained by the use of protective shielding of the neck region [27]. Plaques were present over the whole age spectrum in our patients, starting from <20 years, an age group normally not burdened by plaques [21,28,29]. This indicates early toxic arterial plaque formation and local arterial stiffening as a sequel of radiation therapy in childhood. There was also an increased prevalence of arterial intimal thickening identified in subjects <20 years of age. We have previously reported ultrasound derived arterial intimal thickening among subjects burdened by CVD risk factors starting from age 36 years [12]. Allogenic HSCT survivors showed increased local carotid artery stiffness and increased femoral IMT compared with autologous HSCT survivors.

This study further shows increased prevalence and clustering of CVD risk factors among HSCT survivors compared with the Finnish population of similar age. CIMT has previously been shown to be related with hsCRP, whereas in our sample CIMT was further significantly predicted by cumulative burden of CVD risk factors and treatment with anthracyclines [30,31]. Anthracyclines have previously been linked to adverse cardiac effects, e.g., cardiomyopathies, but preclinical and clinical pilots studies suggest an adverse effect of anthracyclines on arterial endothelial function [32,33]. Confounding mechanisms related with disease, disease severity and different treatment protocols could, however, potentially explain the dose dependence reported in the present study between anthracycline and CIMT.

The Increased CIMT was, contrary to our expectations, not related to TBI exposure, but related mainly with CVD risk factors and metabolic syndrome criteria among HSCT survivors. Importantly, we found no difference between CIMT of healthy controls and HSCT survivors without CVD risk factors. We further showed a significantly steeper increase in CIMT with age among HSCT survivors compared with controls, consistent with CVD related accelerated early arterial ageing among HSCT patients. In combination, these observations suggest that the early prevention of CVD risk factors could potentially prevent cardiovascular disease progression among HSCT patients.

The main limitation of this study is the heterogeneity in diseases and treatment regimes in our relatively small sample including a low number of females that limits a more in-depth stratified analysis evaluating associations related with sex, diseases, treatments or doses, and vascular outcomes. A further major limitation is the differences in CVD risk factor profiles between HSCT patients and our healthy control group without CVD risk factors as our controls serve as a healthy vascular reference. This limits the conclusions that can be drawn from the results. By comparing the prevalence of CVD risk factors with population data, our study setting allows us to conclude that there is a high prevalence of CVD risk factors among the HSCT patients. We can further conclude that HSCT patients have an adverse vascular profile compared to healthy controls as well as with previously reported young adult average Nordic populations [20,21], and we conclude that among HSCT patients the adverse vascular profile is related partly to CVD risk factors and partly to treatment regimes. We can, however, not answer whether the effect of CVD risk factors on the vascular profile is higher among HSCT patients than in a young adult population with a similarly high CVD risk factor distribution. A further limitation is the relatively large drop-out rate in cohort 1, which could predispose to selection bias. There was, however, no significant difference in background factors between participants and non-participants. Significant strengths include a relatively long follow-up time and well characterized background and clinical data as well as the use of recently developed and validated novel methods to assess subclinical changes of the arterial wall. The follow-up time was, however, not long enough for the development of cardiovascular disease events.

In conclusion, we report significant adverse alterations in vascular morphology and function consistent with early arterial ageing in young adult survivors following childhood HSCT. Although plaque formation and arterial stiffening were mainly related with radiation therapy, we found that intima-media thickness was mainly related with clustering of CVD risk factors. This suggests that vascular changes develop through different pathways among HSCT survivors including a direct toxic damage of radiation leading to plaque formation, while CVD risk factor clustering plays a key role in premature and early intima-media layer thickening. The results suggest that early interventions to modify CVD risk factors potentially could improve long-term cardiovascular outcomes in this high-risk population.

## Figures and Tables

**Figure 1 jcm-09-02208-f001:**
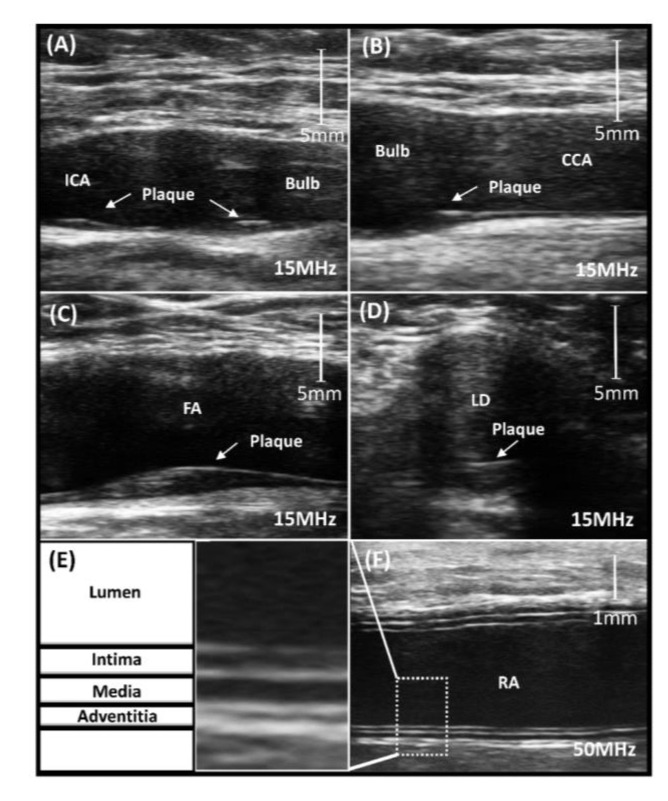
Ultrasound images of bulb and ICA (**A**), and proximal bulb (**B**) of a carotid artery with two plaques (15 MHz). Ultrasound images of a femoral artery with a large hypoechoic dorsal plaque in (**C**) longitudinal and (**D**) transverse imaging planes (15 MHz). Ultrasound image of a radial artery (**F**) with zoomed far wall (**E**) with diffuse intimal thickening seen as a four-line pattern (50 MHz). CCA, common carotid artery; FA, femoral artery; ICA, internal carotid artery; LD, Lumen.

**Figure 2 jcm-09-02208-f002:**
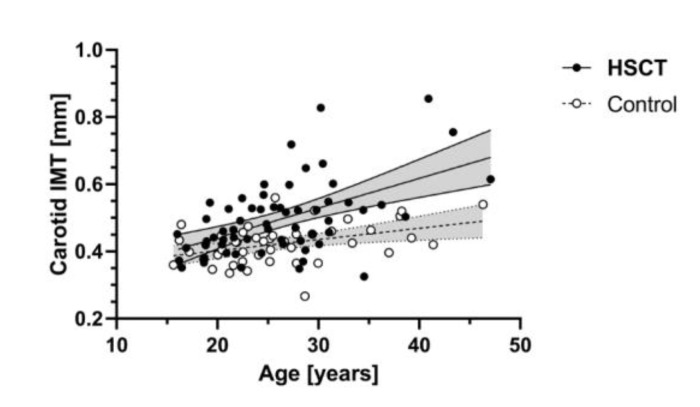
Scatter plots of (A) linear regressions (CI 95% shaded area) of carotid intima-media thickness (CIMT) increase with age among HSCT and controls showing the significant increase in the age-related increase of CIMT in HSCT patients compared to controls.

**Figure 3 jcm-09-02208-f003:**
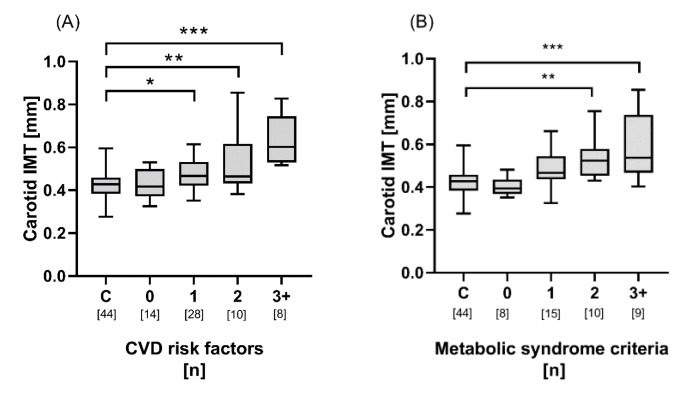
Comparison of carotid artery intima-media thickness (IMT) among controls (C, *n* = 44) and HSCT (*n* = 60) with (**A**) 0, 1, 2, and 3+ of the following cumulative number of cardiovascular disease (CVD) risk factors: hypertension (systolic blood pressure >130 mmHg, diastolic blood pressure >85 mmHg, or antihypertensive medication), HbA1c > 42 mmol/mol, BMI > 30 g/m^2^, anthracycline cumulative exposure > 150 mg/m^2^, and hsCRP >2.5 mg/L and (**B**) 0, 1, 2, and 3+ of the following cumulative number of American Heart Association guideline metabolic syndrome criteria (out of 5) in HSCT cohort 1 (*n* = 42): high waist circumference (>102 cm in men and 88 cm in women); high plasma triglycerides (>1.7 mmol/L); or treatment for hypertriglyceridemia, low plasma high-density lipoproteins (<1.03 mmol/L in men and <1.3 mmol/L in women), hypertension (systolic blood pressure >130 mmHg, diastolic blood pressure >85 mmHg, or antihypertensive medication), and high fasting plasma glucose (>6.1 mmol/L or glucose lowering medication). * *p* < 0.05; ** *p* < 0.01; *** *p* < 0.001.

**Table 1 jcm-09-02208-t001:** Study subject characteristics at follow-up.

	HSCT (*n* = 62)		Controls (*n* = 44)		
	Mean	SD	Mean	SD	
	Median	Q1; Q3	Median	Q1; Q3	
	*n*	%	*n*	%	*p*-Value
Age [years]	25.9	21.1; 30.1	25.2	21.6; 29.9	0.914
Sex [Female]	11	18%	11	25%	0.467
Height [cm]	165.5	10.8	177.6	8.8	<0.001
Weight [kg]	64.6	19.0	76.5	11.9	<0.001
Body surface area [m2]	1.71	0.30	1.94	0.19	<0.001
BMI [kg/m2]	23.1	4.9	24.2	3.2	0.208
Waist-hip ratio ^a^	0.94	0.08	0.87	0.06	<0.001
Heart rate [1/min]	77.3	16.5	62.0	12.1	<0.001
Systolic blood pressure [mmHg]	124.6	13.4	120.9	10.1	0.114
Diastolic blood pressure [mmHg]	71.1	10.1	67.8	6.7	0.047
Overweight (BMI 25+) [*n*]	18	29%	15	34%	0.671
Obese (BMI 30+) [*n*]	7	11%	2	5%	0.300
*Cardiovascular disease risk factors*					
Fat percentage [%]	29.3	8.7	-	-	
Hypercholesterolemia [*n*]	21	34%	1	2%	<0.001
LDL [mmol/L]	3.22	0.89	-	-	
Triglycerides [mmol/L]	1.28	0.86; 2.00	-	-	
hsCRP [mg/L]	1.55	0.72; 3.77	-	-	
Hypertension [*n*}	21	34%			
Diabetes [*n*]	5	8%	0	0%	0.075
HbA1c [mmol/mol]	35	32; 38	-	-	
Metabolic syndrome [*n*] ^a^	9	21%	0	0%	0.021
Blood pressure medication [*n*]	9	15%	0	0%	0.010
Statin medication [*n*]	6	10%	0	0%	0.040
Previous or current smoking [*n*]	20	32%	0	0%	<0.001
Current Smoking [*n*]	10	16%	0	0%	0.005

Anthropometric measures and cardiovascular disease risk factors among study subjects. Laboratory tests are not available for controls. ^a^ Data available for HSCT cohort 1 only (*n* = 43). BMI, Body mass index, LDL, Low-density lipoprotein, HbA1c, Glycated Hemoglobin.

**Table 2 jcm-09-02208-t002:** Comparison of vascular parameters between HSCT and controls.

	Unadjusted					Adjusted for Age and BSA	
	HSCT		Control			HSCT-Control	
	Mean	SD	Mean	SD	*p*-Value	Δ Mean	CI95%
	*n* = 62		*n* = 44				
Common carotid artery							
LD [mm]	5.39	0.55	5.47	0.50	0.410	0.13	−0.04; 0.31
IMT [mm]	0.49	0.11	0.42	0.06	<0.001	0.07 ^b^	0.04; 0.11
Radial artery							
LD [mm]	1.85	0.37	1.96	0.36	0.119	0.05	−0.08; 0.19
IMT [mm]	0.16	0.03	0.16	0.03	0.355	0.014 ^a^	0.00; 0.03
AT [mm]	0.07	0.02	0.07	0.03	0.597	0.00	−0.01; 0.01
Brachial artery							
LD [mm]	3.47	0.74	3.63	0.61	0.224	0.16	−0.05; 0.36
IMT [mm]	0.18	0.05	0.16	0.03	0.006	0.03 ^b^	0.02; 0.05
AT [mm]	0.13	0.03	0.12	0.02	0.275	0.01	0.00; 0.02
Femoral artery							
LD [mm]	6.71	1.01	7.84	1.00	<0.001	−0.65 ^a^	−1.02; −0.28
IMT [mm]	0.37	0.09	0.33	0.07	0.021	0.04 ^a^	0.01; 0.07
AT [mm]	0.26	0.08	0.25	0.05	0.177	0.01	−0.02; 0.04
Intimal thickening	*n*	%	*n*	%	*p*-value		
Any arteries [n]	18	31%	1	2%	<0.001	-	-
Femoral artery [n]	8	14%	0	0%	0.019	-	-
Radial artery [n]	9	15%	1	2%	0.042	-	-
Plaques	*n*	%	*n*	%	*p*-value		
Any plaques [n]	18	30%	2	5%	0.001	-	-
Carotid Plaque [n]	15	24%	2	5%	0.007	-	-
Femoral Plaque [n]	8	13%	0	0%	0.020	-	-
Arterial stiffness	Mean	SD	Mean	SD	*p*-value	ΔMean	CI95%
CBSI	5.5	1.7	4.4	1.4	<0.001	1.45 ^c^	0.85; 2.05
CDC [%/10 mmHg]	4.4	1.5	5.9	1.9	<0.001	−1.78 ^c^	−2.38; −1.19
Carotid-femoral PWV [m/s]	8.7	1.4	8.5	1.4	0.656	0.05	−0.68; 0.78
Carotid-radial PWV [m/s]	9.5	1.9	9	1.5	0.286	0.03	−1.02; 1.07

Comparison of vascular parameters between HSCT and controls both unadjusted and adjusted for age and BSA. PWV data was available for HSCT cohort 1 only. AT, Adventitia thickness; BSA, Body-surface area; CBSI, Carotid β-stiffness index; CDC, Carotid distensibility coefficient; CI95, 95% confidence interval; IMT, Intima-media thickness; LD, Lumen diameter; PWV, Pulse-wave velocity. ^a^ significant at 0.05-level. ^b^ significant at 0.01-level. ^c^ significant at 0.001-level.

**Table 3 jcm-09-02208-t003:** ANCOVA model predicting carotid intima-media thickness among HSCT.

Dependent Variable	n	R^2^	Model *p*-Value
Carotid intima-media thickness [µm]	62	0.597	<0.001
Independent variables	β	CI95%	*p*-value
Constant	176.8	79.6; 274.0	0.584
Age [years]	8.5	5.3; 11.8	<0.001
BMI [kg/m^2^]			
<25	-	-	-
25–30	−15.1	−70.1; 39.9	0.584
>30	85.2	14.2; 156.1	0.020
HbA1c > 42 mmol/mol [0 = no, 1 = yes]	86.6	24.3; 148.8	0.007
Hypertension [0 = no, 1 = yes]	50.5	1.6; 99.4	0.043
hsCRP > 2.5mg/L	56.3	11.9;100.7	0.014
Anthracycline cum. dose/BSA [mg/m^2^]			
0. No Anthracyclines	-	-	-
1. 0–150 mg	55.38	−1.5; 112.2	0.056
2. >150 mg	80.0	28.2; 131.7	0.003

ANCOVA-model predicting carotid intima-media thickness among HSCT. Inclusion of LDL/HDL, triglycerides, smoking, total body irradiation, or exposure to cyclophosphamides did not affect the fit of the model. BMI, Body mass index; HbA1c, Glycated Hemoglobin; hsCRP, high sensitivity C-reactive protein; CI95, 95% confidence interval.

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
