# Peer review of "Early Arterial Intimal Thickening and Plaque Is Related with Treatment Regime and Cardiovascular Disease Risk Factors in Young Adults Following Childhood Hematopoietic Stem Cell Transplantation"

_jcm, 2020, doi:10.3390/jcm9072208_

Round 1
Reviewer 1 Report
The authors investigated early intimal thickening and plaque in young adults following childhood hematopoietic stem cell transplantation. This study seems to be interesting. As a reviewer, I have one request.
#1 The control subjects were recruited among hospital employees. Thus, the present control group was well-controlled and they had extremely lower frequencies of cardiovascular disease risk factors. The authors commented it in the "Limitation" section, however, they should strengthen it in the "limitation" section.
Author Response
We thank the reviewer for the comments. Please find the point-by-point responses in the attached *.pdf.

Reviewer 2 Report
Abstract: good as it is.
Study subjects and methods:
- p2L74: the reference number from the local ethics committee is missing
Results:
- p5L205: the authors present in the text:"similar prevalence of hypercholesterolemia (33.9 vs. 28.6%, 205 p=0.353)" whereas table I Shows a signigficant differnece in hypercholesterolemia. PLease clarif
Discussion:
- good and thorough
Author Response
We thank the reviewer for the comments. Please find the point-by-point responses in the attached *.pdf.

This manuscript is a resubmission of an earlier submission. The following is a list of the peer review reports and author responses from that submission.
Round 1
Reviewer 1 Report
The authors investigated the presences of early arterial intimal thickening and plaque in patients who underwent hematopoietic stem cell transplantation (HSCT), comparing with those in the control population. They showed that the presences of coronary risk factors and/or metabolic risk factors rather than HSCT-related factors influence the development of carotid atherosclerosis. This study included the detail examinations of atherosclerosis and had some difficulties in collecting the studied patients, however, there were some problems to be solved.
#1 In the present study, the studied patients had high frequencies of coronary risk factors and/or metabolic risk factors. On the other hand, the controlled population had the extremely lower frequencies of them. According to the extreme differences in the studied and control population, the coronary risk factors and/or metabolic risk factors rather than irradiation may affect the development of atherosclerosis. The authors should align two groups regarding not only age and sex but also coronary risk factors and/or metabolic risk factors, as much as possible.
#2 As the authors mentioned in the “Limitation” section, the studied population had the heterogeneity of diseases and treatments. The readers will have interest in the relationship between the development of atherosclerosis and treatments such as autologous or allogenic HSCT, the presence or absence of total body irradiation (TBI), and the total dose of irradiation.
#3 In the “discussion” section, the authors had better provide any possible mechanisms responsible for anthracyclines-induced atherosclerosis, as shown in Table 3.
Author Response
We appreciate the interest that the reviewers have taken in our manuscript and the constructive criticism they have given. We have addressed the major concerns of the reviewers and included a point-by-point response to the reviewers’ comments. Please find the attachment.

Reviewer 2 Report
In the present work (Early arterial intimal thickening and plaque is related with treatment regime and cardiovascular disease risk factors in young adults following childhood hematopoietic stem cell transplantation), authors expose the prevalence of cardiovascular disease risk factors (CVDRF) in survivors of childhood hematopoietic stem cell transplantation (HSCT) and how this CVDRF favor the presence of arterial intimal thickening. As authors say, although relationship between arterial intimal thickening and CVDRF is well described in general population, this relationship has not been well described in this population.
Authors include HSCT survivor patients and healthy indivivulas as a control group, to compare the prevalence of CVDRF among this groups. Moreover they try to explain which clinical variables can lead to different profiles of vascular morphology and function alteration.
The present work expose a new and interesting idea. Although working hypothesis is well planned, there are some major concerns about how it is carried out and more data should be included to support the working hypothesis.
Major concerns:
1) The first cohort of HSCT patients (patients treated with myeloablative HSCT) is composed initially of 85 patients, but only 43 are included for the posterior analysis. It implies a 49% loss of patients, which may lead to a selection bias. At least, authors should include it as a study limitation.
2) Control group is formed by healthy individual. Although they are sex and age matched with patient group, they are clearly different from this patient group and thus, no direct comparison should be made between groups. As it can be seen in table 1, although no differences are found in BMI, controls are higher, have more weight and more waist-hip ratio. Also, in patient group there is more prevalence of metabolic syndrome, diabetes, smoking, hypercholesterolemia and cardiovascular treatments. Moreover, in control group no laboratory tests are done to compare lipid profile between groups. In this sense, authors should include a more balanced control group, with a more similar distribution of cardiovascular disease risk factors.
3) Authors conclude that “among HSCT recipients (...) arterial walls were thicker, stiffer and presented with more plaques and intimal thickening consistent with early vascular ageing”. Authors can not conclude this, since patients and controls are not balanced. Moreover, comparing HSCT patients with 0 CVDRF and controls, no differences are found in arterial intimal thickening.
Minor concerns:
a) Table 1 compares control and patient group, but no significance test p values are reported for the comparison of cardiovascular disease risk factors. Thus a significance test p value should be included for variables in rows 13th to 24th.
b) Authors conclude that HSCT survivors have more prevalence of CVDRF than controls, but healthy individuals are selected as control group, instead of a randomly selected population, so this conclusion can not be concluded from this results.
Author Response
We appreciate the interest that the reviewers have taken in our manuscript and the constructive criticism they have given. We have addressed the major concerns of the reviewers and included a point-by-point response to the reviewers’ comments. Please see the attachment.

Round 2
Reviewer 1 Report
The authors revised their manuscript appropriately based on the reviewers' comments. Now I have no requests and questions regarding the revised one.
Reviewer 2 Report
Response to authors:
Major concerns:
2) Control group is formed by healthy individual (…) and thus, no direct comparison should be made between groups (…).
Authors correctly affirm that it has been reported a higher prevalence of cardiovascular risk factors among HSCT patients. In this sense, they intend to study the prevalence of cardiovascular risk factors in this population and its possible implication the development of arterial intimal thickening.
But this work does not have a proper control group, since it is not a population based control group, it is selected from a previously known healthy population. Because that, neither a more prevalence of cardiovascular risk factors among HSCT patients, compared with general population, nor the influence of HSCT and cardiovascular risk factors, in the development of arterial intimal thickening can be inferred from this paper.
Moreover, authors keep affirming that HSCT patients have an increased prevalence of cardiovascular risk factors and that this increased prevalence can lead to an adverse vascular profile (lines 340-344).